# Transcriptomic Profiling of Human Limbus-Derived Stromal/Mesenchymal Stem Cells—Novel Mechanistic Insights into the Pathways Involved in Corneal Wound Healing

**DOI:** 10.3390/ijms23158226

**Published:** 2022-07-26

**Authors:** Fatemeh Tavakkoli, Mukesh Damala, Madhuri Amulya Koduri, Abhilash Gangadharan, Amit K. Rai, Debasis Dash, Sayan Basu, Vivek Singh

**Affiliations:** 1Prof. Brien Holden Eye Research Center, LV Prasad Eye Institute, Hyderabad 500034, India; fatemehtavakkoli88@gmail.com (F.T.); mukeshdamala@lvpei.org (M.D.); madhurikamulya333@gmail.com (M.A.K.); sayanbasu@lvpei.org (S.B.); 2Center for Genetic Disorders, Banaras Hindu University, Varanasi 221005, India; akrai10@gmail.com; 3School of Life Sciences, University of Hyderabad, Hyderabad 500046, India; 4Manipal Academy of Higher Education, Manipal 576104, India; 5CSIR-Institute of Genomics and Integrative Biology, Mathura Road Campus, New Delhi 110025, India; gangutalk@gmail.com (A.G.); ddash@igib.in (D.D.); 6Center for Ocular Regeneration (CORE), Prof. Brien Holden Eye Research Center, LV Prasad Eye Institute, Hyderabad 500034, India

**Keywords:** cornea, limbus, mesenchymal stem cells, wound healing, stromal cells, RNA sequencing, transcriptome, ocular surface, tissue remodeling, regeneration

## Abstract

Limbus-derived stromal/mesenchymal stem cells (LMSCs) are vital for corneal homeostasis and wound healing. However, despite multiple pre-clinical and clinical studies reporting the potency of LMSCs in avoiding inflammation and scarring during corneal wound healing, the molecular basis for the ability of LMSCs remains unknown. This study aimed to uncover the factors and pathways involved in LMSC-mediated corneal wound healing by employing RNA-Sequencing (RNA-Seq) in human LMSCs for the first time. We characterized the cultured LMSCs at the stages of initiation (LMSC−P0) and pure population (LMSC−P3) and subjected them to RNA-Seq to identify the differentially expressed genes (DEGs) in comparison to native limbus and cornea, and scleral tissues. Of the 28,000 genes detected, 7800 DEGs were subjected to pathway-specific enrichment Gene Ontology (GO) analysis. These DEGs were involved in Wnt, TGF-β signaling pathways, and 16 other biological processes, including apoptosis, cell motility, tissue remodeling, and stem cell maintenance, etc. Two hundred fifty-four genes were related to wound healing pathways. *COL5A1* (11.81 ± 0.48) and *TIMP1* (20.44 ± 0.94) genes were exclusively up-regulated in LMSC−P3. Our findings provide new insights involved in LMSC-mediated corneal wound healing.

## 1. Introduction

The cornea is the transparent and highly specialized tissue located in the anterior portion of the eye. In addition to its function as a protective barrier, the cornea is largely responsible for the transmission of light onto the retina, accounting for two-thirds of the eye’s refractive power [1,2,3]. Anatomically, the cornea is made of three major layers: epithelium, stroma, and endothelium. The epithelium is a 4–6 layered outermost structure made of non-keratinized stratified squamous cells. Stroma is the middle layer comprising ~90% of the corneal thickness and contributing to most of the structural framework. It is made of an extensive network of collagen fibrils with interstitially embedded cells called keratocytes, and proteoglycans such as lumican, keratocan, and decorin. The stroma is followed by endothelium, the innermost layer. Endothelium is majorly responsible for the maintenance of stromal dehydration via pumping out excess water/fluids, which in turn prevents corneal edema and resultant opacity. Any damage to one or more of these layers due to extrinsic or intrinsic factors affects the cornea’s transparency [4], a crucial factor for optimal vision [5]. Between the transparent cornea and opaque sclera is the transitional zone, limbus. This acts as a storehouse of the stem cells required for corneal homeostasis and regeneration [6,7,8].

Corneal epithelium, the outermost surface, is subjected to microscopic wear and tear, which requires constant renewal of the lost cells or damaged tissue. The maintenance of the corneal epithelium and stroma relies on the populations of limbal epithelial stem cells (LESCs) and limbal stromal stem cells. Located at the base of limbal crypts [9,10], the LESCs interact with the underlying cells of the limbal stroma [11] through the interruptions in the basement membrane. Limbal stroma is a highly vascularized tissue [12] that has a mixed population of fibroblast-like cells, melanocytes, myofibroblasts, and nerve cells, as well as transmigrating immune cells such as dendritic cells, lymphocytes, mast cells, and macrophages. Derived from the neural crest [13], these limbus-derived stromal cells are multipotent [14,15,16,17] mesenchymal stem cells that conform to the ICST (International Society for Cellular Therapy) criteria [18], demonstrating their trilineage differentiation potential [19,20]. Multiple studies have shown that these LMSCs can also trans-differentiate to keratocyte lineage [14,20,21,22,23] and epithelium [24,25]. They support and regulate the plasticity and niche of the LESCs towards the restoration of the impaired limbal niche and corneal wound healing [12,26,27,28,29,30,31]. The migration of both LESCs and LMSCs to the site of injury and the subsequent combined repair mechanisms are responsible for the maintenance of the stem cell functions and restoration of corneal transparency, a pre-requisite for optimal vision [22,32,33,34,35,36,37,38,39]. 

In case of injuries involving the stroma, the native keratocytes transdifferentiate into fibroblasts and then myofibroblast cells, facilitating the migration and healing of the damage in the corneal stroma. However, this wound healing mechanism is undesirable, as it leads to fibrosis causing corneal haze and scar formation. This obscures the visual pathway leading to partial or complete visual impairment [40,41,42].

The most common surgical means of treating an injured, melted, or the perforated cornea is partially or completely replacing it [43]. The currently available modes of treating these pathologies are often challenged by risks such as graft failure or rejections, inflammatory responses, long-term follow–up, and the inadequate supply of donor corneas [5]. Besides, the current procedures do not offer longevity and unaffected or optimal visual acuity post-transplantation [44,45,46,47,48].

Limbal stromal stem cells were earlier reported for their safety and efficacy [8,9] in preventing corneal scars and the regeneration of corneal stroma. However, the underlying molecular mechanisms behind the stem cell-based for cornel regeneration are not well studied.

In the current study, we have attempted to uncover the regulatory pathways in-volved in LMSC-mediated corneal wound healing. We examined the human LMSCs in comparison to the native tissues of the limbus and cornea using the RNA-Seq. Scleral tissue was used as a control. The detected DEGs were subjected to pathways through the Gene Ontology studies to obtain insights into various biological/signaling pathways. The differences in the frequency distribution of fold-change values of pathway-specific genes were compared to that of all the other genes in the transcriptome. The exclusively up-regulated genes in the corneal wound healing process were checked for their known and probable interactions with other genes through STRING analysis and validated through qRT-PCR. Our data provides molecular/mechanistic insights into corneal wound healing mediated by LMSCs.

## 2. Results

### 2.1. Expansion of Limbal Stem Cells in Culture

Both limbal epithelial (round cells which grow as a layer) and limbal stromal (spindle-shaped and individual) cell populations were obtained from the explants in the initial (Figure 1A) and late stages (Figure 1B,C) of the primary culture (>80% confluence, days 8–10). Further subcultures were observed to show the elevated number of stromal cells and the gradual decrease in the epithelial cells from passages P1 (Figure 1D) to passages P2 (Figure 1E). A pure population of the limbal stromal cells without any presence of the epithelial cells was derived in passage P3. This population of stromal cells also featured the presence of few myofibroblasts in their undifferentiated state, (Figure 1F; orange arrow), dendritic cells (Figure 1F; black arrow), and few quiescent fibroblastic cells (Figure 1F; white arrow), similar to the native corneal stroma.

### 2.2. Cell Type Biomarker Changes during Culture Passages

#### 2.2.1. Stem Cell and Ocular Biomarkers

The immunostaining analysis has revealed a similar pattern of the expression (positive) of the stem cell (ABCG2, p63-α) and ocular biomarkers (PAX6) at both LMSC−P0 and LMSC−P3 stages of the culture. However, ABCB5 was found to be expressed in a low number of cells in P0 relative to P3. Additionally, the number of cells positive for p63-α were high in P0 relative to P3 (Figure 2). ABCB5 plays a vital role in the differentiation of limbal stem cells and is essential for corneal repair [34].

However, the RT-PCR data have revealed significant up-regulation of *ABCG2* in LMSC−P3 with respect to that of LMSC−P0 and native limbus (Figure 2). *ABCB5* was found to be significantly down-regulated in both LMSC−P0 and LMSC−P3 and *PAX6* was found to be significantly up-regulated relative to limbal tissue. *P63α* was found to be down-regulated by 3-fold in LMSC−P3 relative to native limbal tissue. On the contrary, the level of *P63α* was found to be up-regulated 3-fold in LMSC−P0 relative to the control.

#### 2.2.2. Mesenchymal Stem Cell Markers

The limbal stem cells at LMSC−P0 were found to be positive in relatively low numbers for the mesenchymal stem cell (MSC) biomarkers CD90, CD105, and VIM (Vimentin). However, most of the cells at LMSC−P3 were found to be positive for the above markers (Figure 3). The qRT-PCR analysis revealed a down-regulated expression of markers *CD90*, *VIM* (1-fold), and *CD105* (6-fold) in LMSC−P0 relative to the control, limbal tissue. On the contrary, the levels were found to be up-regulated in LMSC−P3 by 2-fold of *CD105*, 3-fold of *VIM*, and ~20-fold of CD*90* relative to the control (Figure 3).

The transmembrane proteins NCAD (N-cadherin) and ECAD (E-cadherin) were observed to express positively in LMSC−P0 through the immunostaining analysis. However, Ecad was found to be negative in LMSC−P3, and Ncad showed positive expression. The qRT-PCR analysis showed that *NCAD* was found to be up-regulated in both LMSC−P0 (8-fold) and LMSC−P3 (26-fold) compared to the native limbus. *ECAD* was found to be down-regulated in LMSC−P3 (3-fold), while it was found to have an increased expression in LMSC−P0 (3-fold) relative to the control (Figure 3).

### 2.3. Genome Wide Transcriptomics Analysis Using RNA-Seq

#### 2.3.1. Transcriptome Overview Using Principal Components Analysis Plot

To visualize the overall similarities or differences between gene expression patterns in different cell types, the counts were analyzed through Principal Component analysis (PCA). The counts data was subjected to Box-Cox transformation to stabilize the skewness in the data before PCA analysis. This analysis has showed the overall differences in the expression patterns of the samples in terms of the distances between them, which indicates the similarity between their expression profiles. It was found that sclera and cornea clustered together and are quite distant from the other samples. This indicates that the differences in their gene expression is not as heterogeneous (Figure 4A) as compared to the rest of the analytes (Limbus, LMSC−P0, LMSC−P3 and ESC (embryonic stem cell)). 

The limbus tissue and LMSC−P0 were observed to form an isolated cluster away from LMSC−P3 and ESC. The altered transcriptomic signature of LMSC−P3 may possibly be the result of repeated passaging and de-epithelialization. However, it was not very distinct from ESC, indicating possible shared/similar gene expression patterns such as pluripotent nature and dedifferentiation.

#### 2.3.2. Visualizing the Asymmetry in Gene Expression of Various Tissues

Around 28,000 genes were detected via RNA-Seq in all the analytes. Among them, 7800 genes were differentially expressed (either up-regulated or down-regulated) against scleral tissue as a control (Figure 4B). In limbal tissue, a total of 1036 genes were up-regulated and 1093 genes down-regulated. LMSC−P0 had 1570 genes up-regulated and 1838 genes down-regulated, wherein LMSC−P3 774 and 1530 genes were up-regulated and down-regulated, respectively.

The asymmetry in gene expression by cornea, limbus, LMSC−P0, LMSC−P3, and ESC was visualized by plotting their transcriptome through volcanic plots. Volcanic plots provide a visual representation of the DEGs, showing their statistical significance (*p* values) versus the magnitude of change (fold-change). These scattered plots have shown that the transcriptome of corneal tissue had a major proportion of the down-regulated genes (Figure 4C). On the other hand, limbal tissue had a distinct asymmetry, with a major proportion of the genes significantly up-regulated (Figure 4D). LMSC−P0 showed a near symmetry in the plot (Figure 4E), while LMSC−P3 has a smaller proportion of up-regulated genes (Figure 4F). The ESC had shown a large number of down-regulated genes (Figure 4G).

### 2.4. Tissue-Specific Differential Expression and Pathway Enrichment Analysis

The differential expression of the genes that were either specific to a particular type of cell or tissue was analyzed using 5-way Venn diagrams (Figure 4H,I). The information on the number of genes commonly expressed in one or more cells/tissues was obtained. In addition, the number of genes that were either exclusively up-regulated or exclusively down-regulated in one particular type of cell/tissue was also obtained. The number of genes exclusively up-regulated in LMSC−P0 and LMSC−P3 was 459 and 223, respectively, while the exclusively down-regulated ones were 465 and 387, respectively (Figure 4H,I).

Pathway-specific Gene Ontology (GO) enrichment analysis using the Enrichr tool provided insights into various cellular processes and pathways such as apoptosis, cell motility etc., where one or two particular cells/tissues were playing a major role. This was evident from the statistically significant gene expression with respect to the control (sclera). The relative median change indicated the up-regulation or down-regulation of such genes with respect to the basal expression levels of the whole transcriptome (Figure 5A). Few of the prominent or majorly observed cellular processes were plotted against the relative expressions of DEGs specific to each of these processes.

#### 2.4.1. Interpretations from Gene Ontology Enrichment Analysis

A total of 6634 unique genes belonging to 16 relevant biological processes were found to be expressed by the corneal and limbal tissues and cells LMSC−P3, LMSC−P0 and ESC. The relative comparison of DEGs specific to various cell processes expressed by the cells of interest in this study—LMSC−P3 and LMSC−P0 cells—has provided interesting results.

#### 2.4.2. GO Pathway Level Gene Expression Changes with Respect to Whole Transcriptome

The LMSC−P0 was found to have relatively high numbers of genes of the cellular processes apoptosis (*BAX*, *BCL2*, etc.,), mitochondrial biogenesis (*SIRT3*, *CASP8*, etc.,) and its transport (*ATP5F1A*, *BCL2*, etc.,) and respiration (*BID*, *COX10*, etc.,) relative to that of LMSC−P3, and were significantly up-regulated (Figure 5A). Genes of wound healing (*COL5A1*, *TIMP1*, *ANXA1* etc.,), tissue remodeling (*HIF1α*, *NOX2*, *NOTCH4* etc.,), stem cell maintenance (*FOXO1*, *SOX2*, *TP63* etc.,), and cell motility (*MAPK*, *MMP1*, etc.,) were found to be more expressed in high numbers in LMSC−P3 than that of LMSC−P0; however, they were down-regulated with respect to the control (sclera). Genes of epithelial phenotype were found to be strongly down-regulated in LMSC−P3. Genes of the epithelial-to-mesenchymal transition (*SNAI1*, *TWIST1*) were down-regulated in both LMSC−P0 and LMSC−P3. Inflammatory response (*C3*, *CXCL8*, etc.) and tissue remodeling genes (*MMP14*, *MMP2*, *IL15*, etc.) were down-regulated more strongly in LMSC−P3 relative to LMSC−P0. In addition to these processes, the GO analysis revealed the DEGs of various signaling pathways such as Wnt, TGF-β and stem cell pathways (Appendix A).

#### 2.4.3. Genes of Multiple Cell Signaling Pathways

##### Genes Involved in Wound Healing Pathway

Around 254 genes belonging to the wound healing pathway were found to be differentially expressed (GO consortium accession number 0042060). The heat map showing the relative expression of these DEGs (Figure 5B) has shown that more significantly up-regulated genes were expressed by LMSC−P0 cells (relative to sclera), followed by limbal tissue and LMSC−P3. Among these DEGs, 21 genes (*CASP3*, *EPB41L4B*, *AJUBA*, *NFE2*, *EGFR*, *IL24*, *ANXA2*, *HMGCR*, *PRKCQ*, *DSP*, *F3*, *IL1A*, *KLK6*, *UBASH3B*, *RHOC*, *TFPI2*, *ADAM15*, *METAP1*, *RAC2*, *DGKA*, *DCBLD2*) were found be exclusively up-regulated or expressed by LMSC−P0 alone. On the other hand, LMSC−P3 has shown exclusive up-regulation of *TIMP1* and *COL5A1* genes (Figure 5C). These two genes were validated through qRT-PCR with native limbus tissue as control, which revealed that in LMSC−P0, the level of *TIMP1* (16.44 ± 0.87) is up-regulated and *COL5A1* (−9.32 ± 0.53) is down-regulated. In LMSCP3, both the genes were up-regulated: *COL5A1* (11.81 ± 0.48) and *TIMP1* (20.44 ± 0.94) (Figure 5D).

##### Other Signaling Pathways

The AmiGO gene ontology analysis of the total DEGs has revealed that 211 genes playing a role in the Wnt signaling pathway were differentially expressed by cornea, limbus, ESC, LMSC−P3, and LMSC−P0. The relative expression levels of these genes from the RNA-Seq by each cell/tissue were plotted in Appendix A, tabulated in Appendix A. A total of 85genes belonging to the TGF-β signaling pathway were found to be differentially expressed by one or more cells/tissues. Among them, *COL3A1* was found to be exclusively up-regulated in LMSC−P3 alone. 

Of the 23 genes available in the GO database which belong to stem cell pathway, 13 DEGs (Appendix A) were found to be expressed by the cells or tissues analyzed in this study (GO consortium accession CL 00000034). The plot of their relative expression levels has shown (Appendix A) that the majority of genes were found to be significantly down-regulated in LMSC−P0. 

### 2.5. Quantification of Genes Interacting among the Exclusively Up-Regulated Genes in LMSC−P3

String database revealed 17 more genes were interacting with exclusively up-regulated genes, i.e., *COL5A1* and *TIMP1* (Figure 5D). The STRING network was formed with a PPI enrichment *p*-value of <1.0 × 10^−16^ (Figure 6B). The highly co-expressive genes were *COL1A1* and *COL3A1* with the RNA co-expression score of 0.944 (Figure 6C). Biological process involved among these 17 genes were mentioned earlier (Table 1). 

Seventeen genes (*CXCR4*, *HIF1A*, *LUM*, *MMP1*, *MMP3*, *MMP9*, *ACTA2*, *VEGF*, *WNT7A*, *HLADR*, *IL10*, *IL13*, *IL6*, *KERA*, *STAT3*, *TGFB1*, *TIMP2*) were hypothesized through string analysis based on their interactions with exclusively up-regulated genes *TIMP1* and *COL5A1* (Figure 6A). These 17 genes were validated through RT-qPCR. When compared with the native limbal tissue, in LMSC−P0, nine genes were up-regulated, i.e., *CXCR4* (9.23 ± 3.31), *HIF1A* (7.84 ± 0.47), *LUM* (4.06 ± 1.35), *MMP1* (12.83 ± 1.49), *MMP3* (1.88 ± 1.01), *MMP9* (1.88 ± 1.01), *ACTA2/αSMA* (14.58 ± 2.86), *VEGF* (9.65 ± 1.30), and *WNT7A* (6.90 ± 1.30), and 8 genes were down-regulated, i.e., *HLADR* (−2.01 ± 0.11), *IL10* (−7.95 ± 1.09), *IL13* (−1.43 ± 0.97), *IL6* (−2.25 ± 0.46), *KERA* (−1.36 ± 1.42), *STAT3* (−0.72 ± 0.63), *TGFB1* (−2.90 ± 1.29), and *TIMP2* (−5.36 ± 1.37) (Figure 7).

In the pure population of LMSCs, i.e., LMSC−P3, five genes were down-regulated: *IL13* (−3.97 ± 1.06), *MMP3* (−0.96 ± 0.52), *STAT3* (−1.23 ± 0.87), *TGFB1* (−1.04 ± 0.26), and *TIMP2* (−1.70 ± 0.32). Meanwhile, 12 genes were up-regulated: *CXCR4* (4.53 ± 0.36), *HIF1A* (22.51 ± 1.12), *HLADR* (11.22 ± 0.41), *IL10* (4.78 ± 0.43), *IL6* (7.65 ± 1.49), *KERA* (16.45 ± 0.54), *LUM* (8.60 ± 0.92), *MMP1* (13.40 ± 1.13), *MMP9* (3.34 ± 0.38), *ACTA2/αSMA* (30.76 ± 1.70), *VEGF* (26.74 ± 0.76), and *WNT7A* (13.69 ± 1.68) (Figure 7). Among these, 17 genes—*CXCR4*, *HIF1A*, *LUM*, *MMP1*, *MMP9*, *ACTA2/αSMA*, and *VEGF*—were commonly up-regulated, and *IL13*, *TIMP2*, and *TGFB1* were commonly down-regulated in both LMSC−P0 and P3.

## 3. Discussion

Various groups across the globe have attempted to understand the basic biology and the mechanisms involved in the healing or repair of the corneal wounds resulting from trauma [49,50,51] or the regeneration of the corneal epithelium and stroma lost due to the regular wear and tear [30,52]. The epithelial homeostasis is achieved primarily through the LESCs residing in the limbal crypts [53], in which the systematic synthesis and the degradation of the collagens in the stromal extracellular matrix (ECM) released by the native keratocytes helps in the maintenance of the corneal stromal integrity and homeostasis [54]. The interactions between the epithelial and stromal cells affect the repair of the cornea after an injury. Earlier studies have shown that the communication or the interaction between the LESCs and the stromal cells through their cytokines and other secretory molecules is essential for maintaining the corneal integrity [28,37,53] and thereby its transparency. IL-1 and its isoforms (IL-1α and IL-1β), produced by the epithelial cells during corneal injury, promote the production of TNF-α, KGF, and HGF [55,56]. Together with TNF-α, IL-1 also modulates the production of growth factors (PDGF and family) that modulate the chemotaxis and proliferation of corneal fibroblasts [57]. They also enhance the levels of cytokines such as G-CSF, neutrophil-activating peptide, IL-3 precursor, IL-4, IL-6, IL-7, IL-8, IL-9, and IL-17 [58]. These cytokines trigger the entry of inflammatory cells to the site of injury [59,60]. HGF and KGF released by the stromal fibroblasts, along with bFGF, IGF, and EGF, modulate the interactions between epithelial and stromal cells, regulating the migration and differentiation of damaged epithelial cells [61,62,63,64,65]. IL-6, a multifunctional cytokine, modulates the repair of the cornea in many ways. It enhances the epithelial wound closure, and low levels of IL-6 delays the healing [66,67,68]. Additionally, it reduces the levels of IL-1 and TNF-α, lowering inflammation [69]. A study by Samaeekia et al. [37] has shown that the exosomes isolated from the corneal and peripheral limbal MSCs enhance the migration and proliferation of corneal epithelial cells in vitro. The co-culture of corneal epithelial cells and corneal stromal cells has been shown to reduce the levels of pro-inflammatory cytokines and enhance the number of viable epithelial cells following an injury [70].

However, in the cases of corneal injuries (limited to the layers of epithelium and its surface, as well as the stroma), the healing process that follows involves one or more factors such as the native cells, growth factors, genes, cytokines, and antigen-presenting cells and even lipids [71,72,73,74,75]. The healing/repair process could involve just one of the above factors or a cascade of multiple events and reactions based on the site/location and the severity of the wound. Additionally, not all of them can be favorable towards the transparency of cornea. Mechanisms such as corneal fibrosis result in opaque/scarred cornea obscuring the visual pathway. The LMSCs were proven to be one of the promising intervention which could prevent and repair the corneal wound without needing a whole corneal replacement [22,38]. These cells are capable of differentiating into the native keratocyte phenotype [22,23]. Recent studies by Orozco Morales et al. [70], Hertsenberg et al. [76], Weng et al. [77] Chameettachal et al. [78] and Chandru et al. [79] have shown the potential of these cells in healing the cornea both in vitro and in vivo in animal models. However, the underlying mechanisms of how these cells achieve the scarless wound healing is not clearly studied. The current study aimed in uncovering the pathways and genes or other factors involved in the corneal wound repair by the LMSCs.

The LMSCs were isolated from cadaveric donor corneo-limbal rims and cultivated in a GMP-certified clean-room facility. Cells at the primary cultures (P0) where both mesenchymal/stromal stem cells of limbus and LESCs were obtained and cell population at the third passage where a pure population of the limbal mesenchymal/stromal cells were obtained (Figure 1), were subjected to RNA sequencing and immunostaining analysis. The outcomes of these two methods were further validated through the qRT-PCR. The mix population of the cells at primary culture were termed LMSC−P0 and the latter was termed LMSC−P3. The digestion of limbal tissue with collagenase alone and maintenance of low serum levels may possibly have led to the propagation of limbal mesenchymal/stromal cells only. The complete removal of serum may lead to the generation of fibroblastic cells with reduced keratocyte phenotype [80]. Conversely, a low quantity of serum (2%) [22] after digestion with collagenase alone [81] would allow stromal cells to proliferate with gradual loss of epithelial islands in the culture. Cells in both populations were found to express the stem cell ocular biomarkers positively (Figure 2). However, the number of cells positive for collagens and mesenchymal biomarkers was more in LMSC−P3 (Figure 3). The collagens of corneal stromal ECM also followed a similar trend, with more expression in LMSC−P3 (Figure 4).

The principle component analysis plot revealed an altered transcriptomic signature of the LMSC−P3 from the rest of the clusters. Of the 28,000 genes detected, nearly 7800 were found to DEGs from all the samples, with LMSC−P0 having more number of DEGs. The asymmetry of the up-regulated or down-regulated genes visualized through volcanic plots revealed a near symmetry in LMSC−P0 (Figure 5). The gene ontology enrichment revealed 6344 unique genes with functions in more than 16 biological processes (Figure 6 and Appendix A). Genes belonging to signaling pathways such as Wnt (211 DEGs), TGF-β (85 DEGs), stem cell (23 DEGs), and wound healing pathways (254 DEGs) were also obtained (Appendix A). Many studies have proven the anti-inflammatory and immunomodulatory properties of the LMSCs [70,76,77,82]. The findings of the current study also support the anti-inflammatory nature of these cells. The overall genes of the inflammatory response (734) were down-regulated in LMSC−P3 relative to LMSC−P0 (Figure 5A). The pro-fibrotic gene IL-13 (Figure 7), and inflammatory genes C3 and CXCL8 which may lead to corneal neovascularization, etc., were down-regulated in LMSC−P3. Additionally, the anti-inflammatory gene 1L-10 (Figure 7) was up-regulated in LMSC−P3 relative to LMSC−P0.

COL5A1 is a prominent and vital regulator of fibrillogenesis [83], the levels of which were reported to be high during the healing of scars [84,85]. During wound healing, the fibroblasts recruited to the site of injury produce collagens type I and V for extracellular matrix regeneration and restoration of the corneal thickness. In our study, the levels of COL5A1 were found to be higher in LMSC−P3 when compared to LMSC−P0 and native cornea. A similar finding was reported by Ruggiero et al. [86], who have shown that the amount of type V collagen produced by corneal fibroblasts in vitro is higher than that of the native cornea. Moreover, studies by McLaughlin et al. [87] and DeNigris et al. [88] have reported that the altered fibroblasts affect the level of collagen V in vitro. This also justifies and explains the levels of COL5A1 being proportionate to the number of fibroblasts in cells/tissues analyzed, i.e., LMSC−P3, followed by LMSC−P0, cornea, and limbus (Appendix A). The number of fibroblasts was also relatively high in LMSC−P3 compared to LMSC−P0 (Figure 1A–C,F). These findings were similar to the study by Z.H. Guo et al. [89], who provided insights into the molecular mechanisms of differentiation and stemness maintenance by limbal stem cell niche in mice. The collagen genes of corneal stroma are responsible for collagen synthesis, which is predominantly regulated by COL5A1 [90]. The exclusive up-regulation of the COL5A1 by the LMSC−P3 cells evidently shows their ability and makes them an ideal source for repair and regeneration of corneal tissue through collagen fibrillogenesis (Figure 5C,D).

The other gene exclusively up-regulated in LMSC−P3 was TIMP-1, an inhibitor of the matrix metalloproteinases (MMPs), the genes responsible for cleaving collagens. The tissue inhibitor of metalloproteinases (TIMPs), inhibit these MMP genes, highly regulating the corneal ECM. The binding of TIMPs to the MMPs prevents the degradation of the ECM. TIMP-1 inhibits all active MMPs, except membrane type matrixins (MT1-MMP), whereas TIMP-2 inhibits MMP-2, in particular [91,92]. These two groups of genes, i.e., the MMP family and the TIMP family, also a play vital role in the development of cornea [93]. In this study, we have found that TIMP-1, MMP-1, and MMP-9 were found to be up-regulated and that TIMP-2 and MMP-3 were down-regulated in LMSC−P3. A similar trend was observed in the LMSC−P0, except for the levels of MMP-3. Although MMP-9 in LMSC−P3 was up-regulated, the levels of TIMP-1 were much higher in terms of fold-change. Unlike the earlier studies [94,95], the positive correlation between the levels of TGF-β and TIMP1 was also not observed in our study (Figure 5C,D and Figure 7), which did not involve disease condition or the altering of their concentration in culture. Assessing all these genes in a disease condition may provide a better understanding of their respective roles in corneal regeneration.

The exclusively elevated genes on LMSC−P3 interact through various genes and biological processes. The network functional enrichment analysis performed to understand their interactions has revealed a set of interleukins, matrixins, chemokine receptors, and growth factors. Most of these were up-regulated in LMSC−P3 relative to LMSC−P0 (Figure 7). Corneal ECM genes such as Keratocan, Lumican, and SMA were expressed significantly higher in LMSC−P3 relative to LMSC−P0 and native limbus. Lumican and keratocan belong to the SLRP (small leucine-rich proteoglycan) family, which is critical for corneal clarity. They are responsible for the fibrillar organization of the collagens in the ECM of the corneal stroma [96,97]. Both these proteoglycans play a crucial role in corneal wound healing and regulate inflammation by localizing the macrophages to the site of injury and recruiting neutrophils [97]. The levels of lumican and keratocan were reported to decrease during the scarring of cornea [98]. Unlike the studies [99,100] that reported low expression of keratocan by keratocytes in vitro, we observed relatively high levels of keratocan in LMSC−P3. However, when compared to LMSC−P0, where there is no chance of differentiating the expression of keratocyte markers by a diverse set of cell populations and the relatively less number of stromal cells, the high number of stromal cells in LMSC−P3 could attribute to the high levels of keratocan and lumican. The down-regulation of TGF-β could also be attributed to the keratocan levels, as shown by Kawakita et al. [101], with decreasing levels of TGF-β maintaining the levels of keratocan. This indicates the strong keratocyte-like nature of the cells in LMSC−P3 with respect to LMSC−P0. The increased expression of SMA in LMSC−P3 relative to LMSC−P0 could be attributed to the relatively high number of myofibroblastic cells in LMSC−P3 than in LMSC−P0.

We have also found that the expression of VEGFA, a proangiogenic factor, was also significantly high in LMSC−P3. The continuous maintenance of corneal avascularity is important for optimal visual acuity. Angiogenesis is one of the many vital processes in wound healing for the successful repair of damaged tissue. The balance between the proangiogenic and anti-angiogenic factors is mandatory for maintaining corneal avascularity [102]. To assess the levels of angiogeneic factors that regulate the formation of vasculature on corneal surface, certain genes were quantified through qRT-PCR. VEGFA is one of the proangiogenic factors which, besides FGF-2 [102], plays a role in multiple processes such as immune-modulation, epithelialization, collagen deposition, and cell migration [103]. It decreases the duration of wound healing [104]. Although MSCs were reported to potentially lower angiogenesis [105], the surprisingly high expression of the VEGFA in LMSC−P3 (Figure 7) is questionable due to the fact that elevated vasculature over the surface of cornea can potentially affect the visual acuity [106,107]. However, the elevated levels of VEGFA (growth factor-induced or transfected or topically applied) in the wound bed were reported to enhance the wound repair of dermis/skin [108,109,110,111], but not many studies on corneal surface were reported. These elevated levels of VEGFA also contradict decreased expression of MMP9, the factors reported to feedback regulation mechanism [112]. However, other proangiogenic factors such as PDGF and its family (PDGFB, PDGFC, PDGFD, PDGFRA, and PDGFRB) are either unexpressed or down-regulated in LMSC−P3 (Appendix A). Although native corneal epithelial tissue is reported to have detectable levels of VEGFA and sflt-1 [113,114], not much information is available regarding the levels of VEGFA in native limbal tissue. However, the levels of VEGF expression occurs differently in different cells in vitro. The limbal epithelial cells were earlier reported [115] to be anti-angiogenic in nature and the limbal fibroblasts proangiogenic in nature. The corresponding high and low levels of the limbal fibroblasts in LMSC−P3 and LMSC−P0, respectively, could possibly explain the increased levels of VEGFA. However, a contradicting observation was reported much later in a study by Eslani et al. [116], who have shown that the LMSCs are anti-angiogenic. Low levels of VEGFA and high levels of the anti-angiogeneic factors SFLT-1 and PDGF were observed in the secretome of LMSCs. In the current study, determining the levels of SFLT-1, MMP-2, MMP-14, and CTGF genes in the cell populations/tissues tested could have provided a better answer to this conundrum. Further studies to explore/evaluate the levels of VEGF in a corneal wound model treated with LMSCs and monitoring of the progress of healing may be required.

## 4. Materials and Methods

### 4.1. Ethics Approval and Tissue Collection

Human donor corneas (donor age ranged between 18–60 years) were collected from the Ramayamma International Eye Bank (RIEB), LV Prasad Eye Institute, Hyderabad. Overall, 21 therapeutic-grade donor corneas, unutilized for surgical purposes, were used in this study (*n* = 21). The corneas were collected with informed consent and in compliance with the guidelines of the Declaration of Helsinki. Ethical approval was obtained from the Institutional Ethics Committee (Reference number LEC 05-18-081) and the Institutional Committee for Stem Cell Research, of LV Prasad Eye Institute, Hyderabad, India (IC-SCR-Ref No: 08-18-002).

### 4.2. Establishment of Limbal Stem Cell Culture

The tissue processing was done using a stereomicroscope (SZX10, Olympus, Japan) to set up the limbal stromal stem cell culture, as described previously [117], and for total RNA extraction. Briefly, cadaveric corneas were washed with 1X PBS (14190250, Thermo Fisher Scientific, Waltham, MA, USA) fortified with 2× antibiotics (15240062, Thermo Fisher Scientific, Waltham, MA, USA) and were stripped of endothelium and iris. Full thickness limbus was excised in 1× PBS and then fragmented to small pieces in plain DMEM/F12 media (BE04-687F/U1, Lonza, Basel, Switzerland). These tissue fragments were minced for 1–2 min. The dissected limbal tissue is then enzymatically digested by Collagenase type IV (17104019, Thermo Fisher Scientific, Waltham, MA, USA) enzyme (200 IU per one donor rim), added to 1 mL of plain DMEM/F12 media and then incubated for 16 h. The digested tissue is sedimented twice at 1000 rpm/3 min in PBS. The pellet is then suspended in complete media, i.e., DMEM/F12 fortified with 2% Fetal Bovine Serum (SH30084.03, Cytiva Life Sciences, Shrewsbury, MA, USA) and supplemented with human recombinant Epidermal Growth Factor (PHG0311L, Thermo Fisher Scientific, Waltham, MA, USA) and human recombinant Insulin (12585014, Thermo Fisher Scientific, Waltham, MA, USA). This suspension was plated and cultured for 3 generations. Cells upon confluence, at the stages of primary culture (LMSC−P0) and passage 1 (LMSC−P1) and passage 3 (LMSC−P3), were used for analysis.

### 4.3. Immunofluorescence Assay

Cells were grown on the surface of glass coverslips in complete media until confluence. The cells were then fixed with 4% Formaldehyde (30525-89-4- 500G, Fisher Scientific, Bangalore, India) for 10 min and washed twice with 1× PBS before permeabilization with 0.3% Triton-X (T8787-100ML, Sigma-Aldrich, St. Louis, MO, USA) for 20 min and washed thrice. Later, the cells were blocked with 2.5% Bovine Serum Albumin (BSA) (A7096-50G, Sigma-Aldrich, St. Louis, MO, USA) in PBS for one hour at room temperature and incubated overnight at 4 °C with primary antibody (Appendix A) diluted in 1% BSA. This was followed by a wash with PBS thrice for 10 min and incubation with secondary antibody (Appendix A) (diluted in 1% BSA) for 45 min, which was further washed thrice and mounted onto a glass slide using Fluoroshield Mounting Medium with DAPI (ab104139, Abcam, San Francisco, CA, USA) for nuclei counterstaining. Staining of negative controls was done by omitting the primary antibody. Images were documented using an inverted fluorescence microscope (Axio Scope A1, Carl Zeiss AG, Oberkochen, Germany). The biomarker panel of the MSC phenotype was chosen in accordance with the minimal criteria set for multipotent mesenchymal stem cells [18].

### 4.4. RNA Isolation

Total RNA was isolated from tissues (sclera, limbus, and cornea) and limbal stem cells (LMSC−P0 and LMSC−P3) and embryonic stem cell line (ESC) using TRIzol™ reagent (15596018, Thermo Fisher Scientific, Waltham, MA, USA). The spent medium was removed from the 80% confluent cell culture. Cells were then washed with 1× PBS (prepared with DEPC-treated distilled water for RNA isolation) (AM9920, Thermo Fisher Scientific, Waltham, MA, USA), and an appropriate volume of TRIzol™ reagent was added to the cells. The cell lysate was mixed several times through a pipette and transferred to a sterile 1.5 mL micro-centrifuge tube. To the lysate, 0.5 mL of Chloroform (96764, Sisco Research Laboratories, Mumbai, India) was added per every 1mL of TRIzol™ reagent and incubated at room temperature for 15 min. This was followed by centrifugation at 12,000 rpm for 15 min at 4 °C. The aqueous phase was collected in a fresh tube and 1 mL of Isopropanol (Q13825, Thermo Fisher Scientific, Waltham, MA, USA) was added (in equal volumes with TRIzol™ reagent) and incubated at room temperature for 3 min followed by centrifugation at 12,000 rpm for 3 min at 4 °C. RNA pellet was washed with 75% Ethanol (24102, Sigma-Aldrich, St. Louis, MO, USA), air dried, and dissolved in 25 μL of nuclease-free water (AM7020) (volume dependent on size of RNA pellet). RNA was quantified by measuring the absorbance using a spectrophotometer along with the purity evaluation by the ratio of A260/280 (NanoVue™ Plus, 28956058, GE Healthcare Bio-Sciences AB, Chicago, IL, USA). Further confirmation was done through gel electrophoresis, using 1% agarose gel (50004, Lonza, Basel, Switzerland) stained with Ethidium Bromide (93079, Sisco Research Laboratories Private Limited, Mumbai, India). The RNA was treated with DNase I (AM2222, Thermo Fisher Scientific, Waltham, MA, USA) according to manufacturers’ protocol. Briefly, a 30 μL reaction volume containing 30 μg of total cellular RNA, 1× reaction buffer, 6U of DNase I (RNase free), and nuclease-free water. The reaction mix was incubated at 37 °C for 30 min. After incubation, 70 μL DEPC water was added to the reaction mix and the RNA was purified by adding 100 μL TRIzol™ reagent. The RNA was quantified by measuring the absorbance using a spectrophotometer, as previously described, and 1µg each of the RNA from the analytes was used for the RNA-Seq study.

### 4.5. Next Generation RNA Sequencing (RNA-Seq) and Library Preparation

One microgram each of the total RNA from limbus, cornea, sclera, LMSC−P0, LMSC−P3, and embryonic stem cells (ESC) were subjected to RNA sequencing via Illumina platform using the reagents provided in the Illumina® TruSeq® Stranded Total RNA Sample Preparation Ribo-Zero™ kit (RS-122-2201, Illumina, San Diego, CA, USA). The first step involves the removal of ribosomal RNA using Ribo-Zero™ rRNA removal beads provided in the kit. The Ribo-Zero™ rRNA reagent depletes samples of cytoplasmic ribosomal RNA. Following purification, the RNA was fragmented into small pieces by heat digestion using divalent cations (magnesium or zinc) under elevated temperature. The cleaved RNA fragments were copied into first strand cDNA using reverse transcriptase and random primers. This is followed by second strand cDNA synthesis using DNA polymerase I and RNase H. These cDNA fragments then have the addition of a single ‘A’ base and subsequent ligation of the adapter. The products have been purified and enriched with PCR to create the final cDNA library. This sample preparation protocol provides the advantages of (i) strand information on RNA transcript and (ii) library capture of both coding RNA and multiple forms of non-coding RNA. The processed cDNA library of all 6 samples was used for paired end sequencing run (50 × 2 cycles) on the Illumina HiSeq 2500 platform (SY–401–2501, Illumina, San Diego, CA, USA).

#### 4.5.1. Pre-Processing of the RNA-Seq Data for Data Analysis

The Fastq file was obtained from sequencer after trimming the adapter sequences using bcl2fastq program. Fastq data was used for alignment with the hg19 version of the human genome using the TopHat program with options provided as transcript annotation file. The alignment data has been used for guided transcript assembly using the Cufflinks program. After that, we merged transcripts across samples using the Cuffmerge program to make a reference transcript assembly. This merged transcript assembly has been used as a reference to compare for differential gene expression between a pair of samples with the use of Cuffdiff program. The resultant Cuffdiff output file has provided the normalized expression of genes/transcript in the form of counts, and the fold differences converted into log2 values. The details of the reference links of all the software/programs/bioinformatics tools used in analysis of the RNA-Seq data were provided in the Appendix A.

#### 4.5.2. Differential Expression Analysis

The counts obtained for each sample were analyzed by using the EBSeq tool (Appendix A) for differential expression by considering scleral tissue as the control. A list of DEGs was obtained for the tissues with pairwise comparison to sclera, and multiple testing corrections were applied at a False Discovery Rate (FDR) of 0.05 percent. The heatmaps were generated using R software.

#### 4.5.3. Delineating Cell-Specific Gene Expression Patterns and Testing for Pathway Enrichment

To delineate the DEGs in different tissues and cells according to their cellular specific expression, 5-way Venn diagrams were used to find the genes which were exclusively up-regulated and down-regulated. Two types of the gene expression patterns were analyzed. The cell type-specific gene expression to find genes that were exclusively differentially regulated in only one type of the cell/tissue and which may therefore serve as transcriptomic markers to identify the unique cell type. Pairwise overlapping genes that are differentially regulated only in two cell types may indicate a shared functionality between the two cell types. To obtain pathway-level insights into the significance of the exclusively differentially regulated genes, we have conducted pathway enrichment analysis through the Gene Ontology studies using Enrichr tool. This analysis indicates statistically significant groups of genes that are belong to various biological/signaling pathways.

#### 4.5.4. Gene Ontology Pathway-Specific Gene Expression Changes

Using the ontology keywords derived from the pathway enrichments obtained in the Enrichr analysis, the lists of genes specific to the pathway-keywords were obtained from the gene ontology database using the AmiGO tool. These gene lists were used to examine the differences in the frequency distribution of fold-change values of pathway-specific genes as compared to that of all the other genes in the transcriptome. These differences in the distributions were tested for statistical significance using the nonparametric Mann–Whitney–Wilcoxon U test. The median difference between the distributions was used to detect the direction of the shift in expression value. The values of median shift of the pathways across different samples were plotted against the crossed out the values which were not statistically significant (*p* ≤ 0.05).

### 4.6. Reverse Transcriptase PCR

One microgram each of the RNA was reverse transcribed to cDNA using the Superscript™ III First-Strand Synthesis System (18080051, Thermo Fisher Scientific, Waltham, MA, USA) according to the manufacturer’s instructions.

### 4.7. qRT-PCR

Quantitative PCR was performed using 200 ng of cDNA in a final volume of 25 μL reaction mix (K0221, Maxima SYBR Green/ROX qPCR Master Mix (2X), Thermo Fisher Scientific, Waltham, MA, USA) and a 0.2 μM primer concentration. The reaction was carried out using Step One (Applied Biosystems, Life technologies) hardware and software. The reactions were run in triplicates. The gene expression data were normalized to control the variability in expression levels to the geometric mean of the housekeeping gene. The expression level of target genes was represented as a relative expression by using 2^−ΔΔCt^ formula and the graphs were plotted using their Log2 fold-change values. The primer sequences are listed in the Appendix A.

### 4.8. Statistical Analysis

All the experiments were repeated at least thrice with biological triplicates. Statistical analysis was performed using the Graphpad Prism 6 software. The tests employed were Student’s two-tailed *t*-tests, Kruskal–Wallis test, and a nonparametric one-way ANOVA test with *p* values ≤ 0.05 to assess the statistical significance. The results are presented as the mean ± standard deviation.

## 5. Conclusions

In the current study, we report the genes, biological processes, and pathways involved in the limbal stromal/mesenchymal stem cell-mediated corneal wound healing by employing RNA-Sequencing in human LMSCs (LMSC−Passage-0 and LMSC−Passage-3), for the first time. Differential expression of the genes (7800) belonging to the following pathways, namely, apoptosis, cell motility, dedifferentiation, inflammatory response, stem cell maintenance, tissue remodeling, and wound healing pathways, etc., were found. The interactions between the DEGs exclusively up-regulated by the LMSC−P3 in the wound healing pathway (COL5A1, COL1A1 and TIMP1) have revealed the processes involved in tissue remodeling and repair (collagen fibril reorganization, collagen biosynthesis, regulation of the metallopeptidase activity etc.) and the cytokines and other key genes regulating these processes. However, this study is limited by the small sample size, and further comprehensive studies needed to explore and understand all the DEGs and their biological relevance in corneal wound healing. On the whole, the findings of this study provide a brief glimpse into the molecular basis of tissue repair, and the remodeling of the cornea by human LMSCs and the therapeutic potential of this.

## Figures and Tables

**Figure 1 ijms-23-08226-f001:**
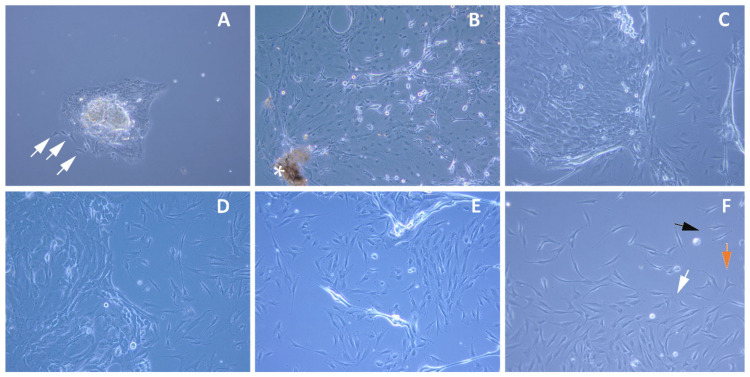
Expansion of the limbal stem cells in culture. Representative images of the limbal stem cells in the primary culture: initiation (**A**) and at confluence (**B**,**C**), passage P1 (**D**), passage P2 (**E**) and passage P3 (**F**). Epithelial cells (round morphology) and stromal/progenitor cells (spindle morphology, indicated with white arrows) derived from the limbal explant (**A**). Gradual increase in limbal stromal cells population and simultaneous fading of limbal epithelial cells (**C**–**E**). Pure population of the limbal stromal cells obtained in passage P3 including dendritic cells ((**F**); black), undifferentiated myofibroblastic cells ((**F**); orange), and quiescent fibroblastic cells ((**F**); white); * Limbal Explant.

**Figure 2 ijms-23-08226-f002:**
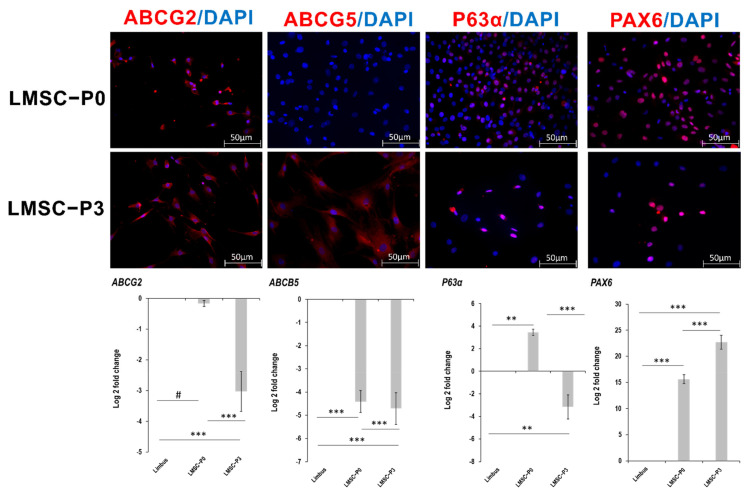
Expression of the stem cell and ocular biomarkers in limbal stem cells. Panel of the representative images of the limbal stem cells showing positive expression of ABCG2, ABCB5, P63-α, and PAX6 (red) in both epithelial (LMSC−P0) and stromal cell (LMSC−P3) populations, counterstained with DAPI (blue). Scale: 50 µm. Level of expression (lower panel) of *ABCG2*, *ABCB5*, *P63α*, and *PAX6* genes quantified using qRT-PCR in limbal epithelial (LMSC−P0) and stromal (LMSC−P3) cells, relative to native limbal tissue (*n* = 5). *P63α* was found to be down-regulated in LMSC−P3 where all the other stem cell genes *ABCG2*, *ABCB5* and *PAX6* were found to follow same pattern of expression in both early and late passages of the culture. The results were plotted as mean log 2-fold change ± SD. The statistical analysis was performed using Kruskal–Wallis one-way ANOVA test. # *p* > 0.05, ** *p* < 0.01, *** *p* < 0.001.

**Figure 3 ijms-23-08226-f003:**
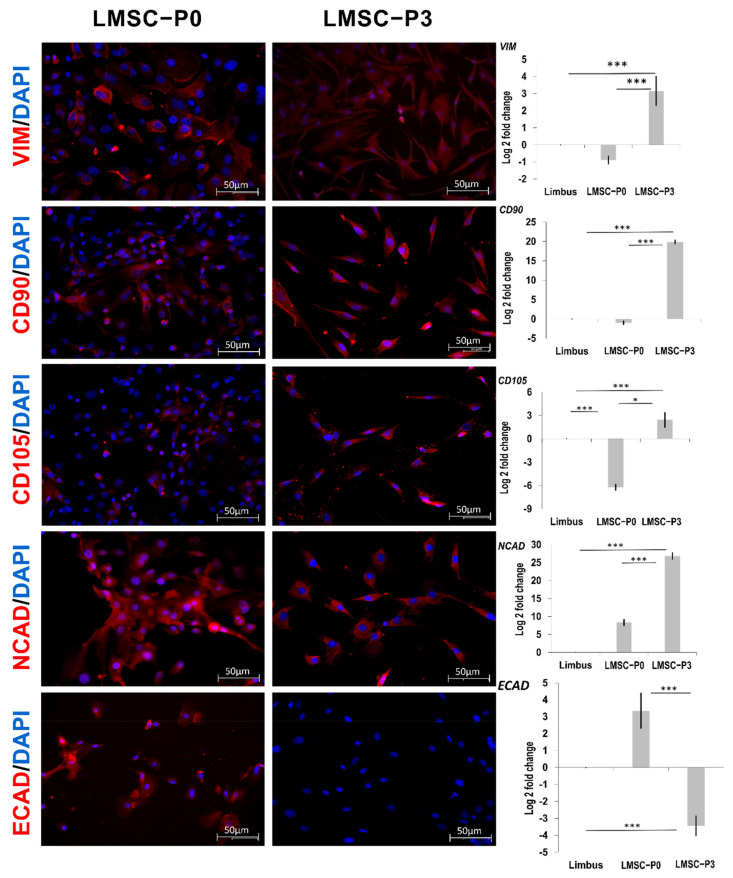
Limbal stem cells showing mesenchymal stem cell biomarkers. Panel of the representative images of the limbal stem cells showing positive expression of Vim (Vimentin), CD90, and CD105 in both LMSC−P0 (*n* = 3) and LMSC−P3 (*n* = 3) populations, counterstained with DAPI (blue). Ncad (N-cadherin) were positive (red) in LMSC−P3 cells and Ecad (E-cadherin) did not show any expression in LMSC−P3. Level of expression of *VIM*, *CD90*, *CD105*, *NCAD*, *ECAD* genes quantified using qRT-PCR in LMSC−P0 and LMSC−P3 relative to native limbal tissue (*n* = 5). Except *ECAD* remaining genes were found to be up-regulated in LMSC−P3 with fold-change ranging between 2 to 20, which were down-regulated in LMSC−P0. Scale: 50 µm. The results were plotted as mean log 2-fold change ± SD. The statistical analysis was performed using Kruskal–Wallis one-way ANOVA test. * *p* < 0.05, *** *p* < 0.001.

**Figure 4 ijms-23-08226-f004:**
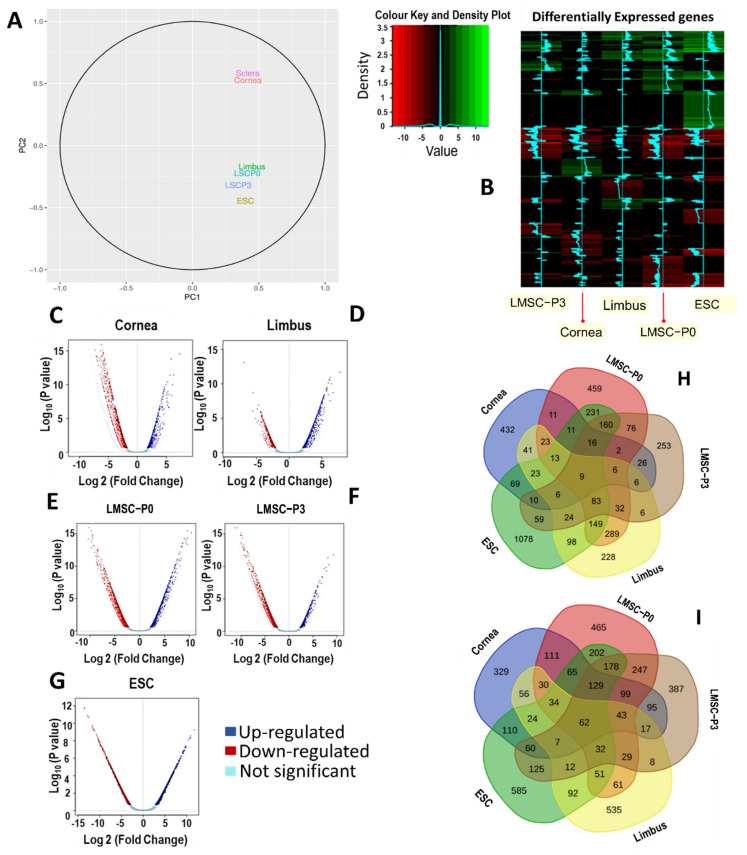
Similarities and asymmetry in the gene expression. (**A**) The count data from all the samples were transformed using Box-Cox transform to compensate for skewness before PCA analysis. The closer proximity of the samples indicates similarity of the expression profiles of those samples. Sclera and cornea were found to show high similarity in whole transcriptome of expression and were clustered together. Similarly, LMSC−P0 and native limbal tissue were in close proximity, indicating similar transcriptomic signature. LMSC−P3 and ESC were found to be further away from one another, indicating an altered or different expression profile relative to the rest of the analytes. (**B**) The heat map representing the DEGs in all 5 samples relative to the control (scleral tissue). The rows indicate the genes and columns indicate the samples (cells or tissues). The color intensity represents the level of changes in expression. All significantly up-regulated genes are indicated in green and all significantly down-regulated genes are indicated in red. *p* < 0.05 was considered to be a statistically significant change in the gene expression. (**C**–**G**) Volcano plots of each cell/tissue samples showing the distribution of genes up-regulated (blue) and down-regulated (red). Majority of the genes in corneal tissue were down-regulated while majority of genes in limbus were up-regulated. The primary culture of limbal stromal cells (LMSC−P0) had nearly equal distribution of the genes that were up-regulated and down-regulated. (**H**,**I**) Tissue-specific differential expression of the genes: Venn diagrams showing the number of genes that are common and exclusively up-regulated (**H**) or exclusively down-regulated (**I**) in cornea, limbus, LMSC−P0, LMSC−P3, and ESC with respect to the scleral tissue (control).

**Figure 5 ijms-23-08226-f005:**
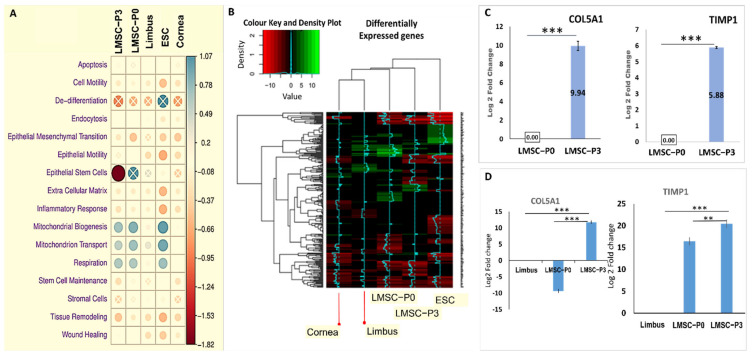
Interpretations from Gene Ontology enrichment analysis. (**A**) Gene ontology pathway-specific analysis: Each row represents the genes belonging to a particular pathway/biological process in the GO database. The dot color/size represents the difference in median expression between genes of a particular pathway and rest of genes in whole transcriptome. A positive value indicates up-regulation (blue) and a negative value indicates down-regulation (red). The difference was tested using Mann–Whitney–Wilcoxon test and statistically insignificant ones were denoted with crosses. Inflammatory response is down-regulated more strongly in P3 versus P0, which may reflect why the use of P3 stage cells does not cause fibrosis in corneal stromal transplants. The stronger down-regulation in ESC may reflect on the immune privilege of embryonic stem cells. The cell motility pathways are active in P3, cornea, and ESC. This is easily explained for the stromal stem cells in P3 and the ESC in terms of their proliferation and migration activity before differentiation, but for cornea may be representative of continuous cell migration required to replace lost corneal tissue. (**B**) The heatmap of 254 genes belonging to wound healing pathway. The rows indicate the genes and columns indicate the samples (cells or tissues). The color intensity represents the level of changes in expression. All significantly up-regulated genes are indicated in green and all significantly down-regulated genes are indicated in red. *p* < 0.05 was considered to be statistically significant change in the gene expression. (**C**,**D**) Genes of wound healing pathway exclusively expressed in LMSC−P3: The levels of COL5A1 and TIMP1 in LMSC−P0 and LMSC−P3 assessed through RNA-Seq (**C**) and qRT-PCR (**D**). The LMSC−P3 has ~10-fold (*n* = 3. *p* < 0.001) high expression of COL5A1, which was down-regulated in LMSC−P0, evident from both techniques. The levels of TIMP1 were ~6-fold high (*n* = 3, *p* < 0.001) and 4-fold higher (*n* = 5, *p* < 0.01) in LMSC−P3, when assessed through RNA-Seq and qRT-PCR respectively. ** *p* < 0.01, *** *p* < 0.001. The results were plotted as mean log 2-fold change ± SD. The statistical analysis was performed using Kruskal–Wallis one-way ANOVA test for qRT-PCR and two-tailed T test for RNA-Seq analysis.

**Figure 6 ijms-23-08226-f006:**
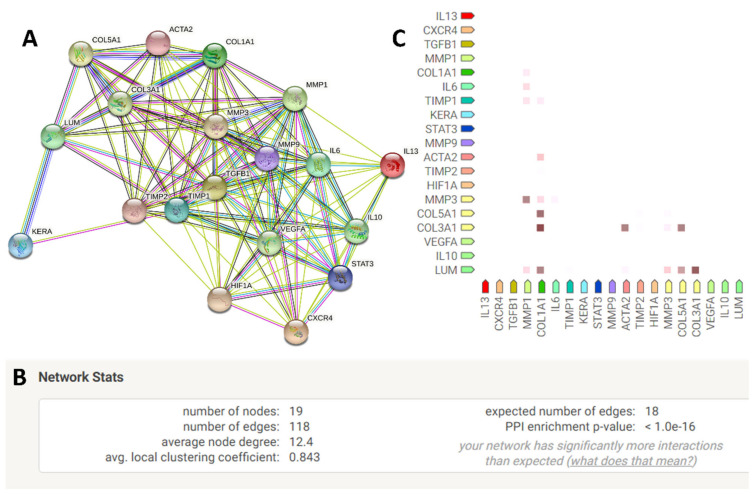
The network generated from the gene interactions involved between COL5A1 and TIMP1 (**A**). The network stats of the network in (**A**) showing the significance value of <1.0 × 10^−16^ (**B**). The RNA co-expression analysis from STRING software shows for the above network shows the COL3A1 and COL1A1 are highly co-expressing in the homeostatic conditions (**C**).

**Figure 7 ijms-23-08226-f007:**
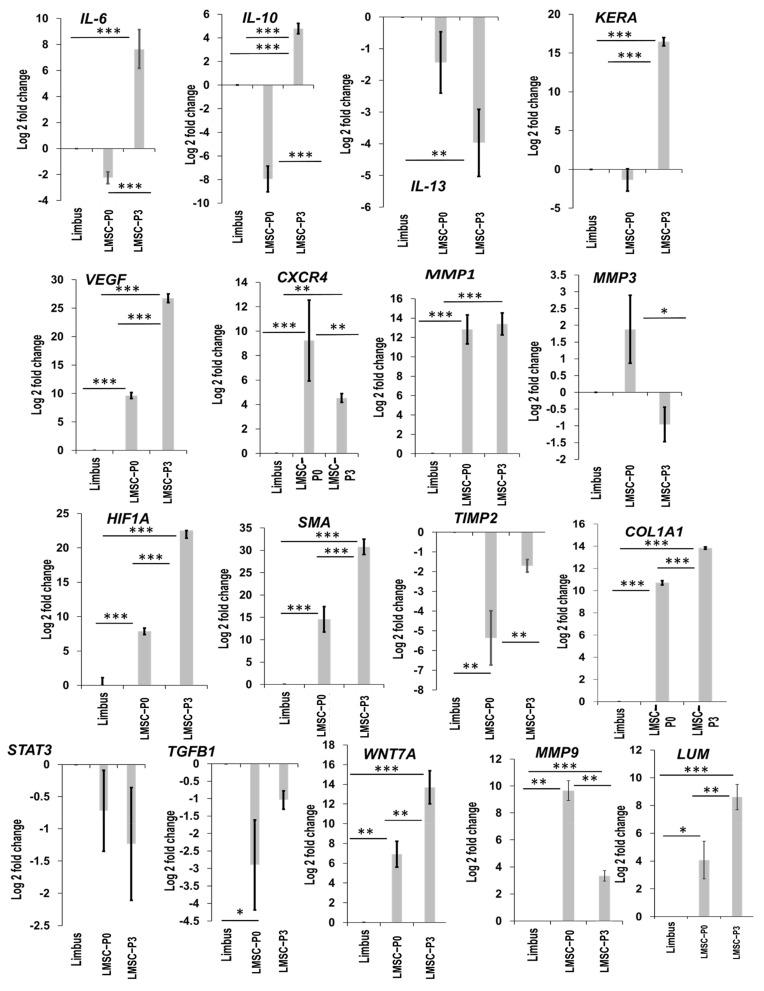
Validating the levels of DEGs. Level of expression of the differentially expressed genes validated through qRT-PCR. The levels of DEGs in limbal epithelial (LMSC−P0) and stromal (LMSC−P3) cells was quantified relative to native limbal tissue (*n* = 5). The results were plotted as mean log 2-fold change ± SD. The statistical analysis was performed using Kruskal–Wallis one-way ANOVA test. * *p* < 0.05, ** *p* < 0.01, *** *p* < 0.001.

**Table 1 ijms-23-08226-t001:** The above table shows the genes playing roles in specific biological processes from the network generated between COL5A1 and TIMP1 gene interactions, which has a significant role in corneal wound healing.

Gene Ontology ID	Biological Process	Genes Involved	False Discovery Ratio
GO:0032964	Collagen biosynthetic process	COL5A1, COL1A1	0.0028
GO:1905048	Regulation of metallopeptidase activity	TIMP1, TIMP2, STAT3	0.00013
GO:0070102	Interleukin-6-mediated signaling pathway	IL-6, STAT3,	0.0061
GO:0030199	Collagen fibril organization	COL5A1, COL1A1, COL3A1, LUM	3.5 × 10^−5^
GO:0035633	Maintenance of blood-brain barrier	VEGF, IL-6	0.0171
GO:0048661	Positive regulation of smooth muscle cell proliferation	MMP9, IL-6, IL-13, IL-10	0.00021
GO:0042060	Wound healing	COL5A1, COL1A1, COL3A1, TIMP1, HIF1A, VEGFA, IL-6, TGFB-1	3.4 × 10^−6^
GO:0060485	Mesenchyme development	ACTA2/SMA, TGFB1, HIF1A	0.0299

## Data Availability

Data available upon request from the corresponding author, due to the Institutional policies.

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
