# Peer review of "Transcriptomic Profiling of Human Limbus-Derived Stromal/Mesenchymal Stem Cells—Novel Mechanistic Insights into the Pathways Involved in Corneal Wound Healing"

_ijms, 2022, doi:10.3390/ijms23158226_

Round 1
Reviewer 1 Report
This study investigated limbal stem cells using RNA-seq to identify differences in gene expression based on (2) timepoints. They identified a number of differences in gene expression at the different timepoints in isolated cells compared to limbus tissue. Overall, the study design and data are interesting. Suggestions are made to improve the quality of the manuscript.
-The authors show that the limbal culture at P0 is a mixed cell population (epithelial + keratocyte-like cells). The gene expression data seems to confirm this showing increased keratocan and lumican at P3. The results should be described to specify that the P0 data is not distinguishing between epi and stromal cell types, and thus is difficult to interpret. Also, elaborating on the expression of keratocyte-specific genes in their dataset at P3 would be beneficial.
-Please add mention to the role of ABCB5 in cornea/stem cell markers (line 121-122).
-Please list the sample size used in the appropriate figure legends. Also, specify if the graphs shown are reporting mean +/- stdev or stderror. The statistical test used should also be listed in the figure legend.
-The authors focus on VEGF and COL5 as genes that are differentially expressed at P3 compared to P0. These findings are unexpected. Can the authors elaborate in the discussion regarding why this might be seen. Have there been comparisons between isolated corneal epithelial cells and keratocytes/fibroblasts in vitro?
-Minor: Change 'viz' to 'e.g.,'
Author Response
Point 1: The authors show that the limbal culture at P0 is a mixed cell population (epithelial + keratocyte-like cells). The gene expression data seems to confirm this showing increased keratocan and lumican at P3. The results should be described to specify that the P0 data is not distinguishing between epi and stromal cell types, and thus is difficult to interpret. Also, elaborating on the expression of keratocyte-specific genes in their dataset at P3 would be beneficial.
Response 1: Thank you for your suggestion. The points suggested are amended in the Discussion, paragraph 7 in lines 480-497, with relevant references (102-107) as
“Lumican and keratocan belong to the SLRP (small leucine-rich proteoglycan) family, which are critical for corneal clarity. They are responsible for the fibrillar organization of the collagens in the ECM of the corneal stroma [102,103]. Both these proteoglycans play a crucial role in corneal wound healing and regulate inflammation by localizing the macrophages to the site of injury and recruiting neutrophils [103]. The levels of lumican and keratocan were reported to decrease during the scarring of cornea [104]. Unlike the studies [105,106] that reported low expression of keratocan by keratocytes in vitro, we observed relatively high levels of keratocan in LMSC-P3. However, when compared to LMSC-P0, where there is no chance of differentiating the expression of keratocyte markers by a diverse set of cell populations and the relatively less number of stromal cells; the high number of stromal cells in LMSC-P3 could attribute to the high levels of keratocan and lumican. The downregulation of TGF-β could also be attributed to the keratocan levels, as shown by Kawakita et al. [107], that decreasing the levels of TGF-β would maintain the levels of keratocan. This indicates the high keratocyte-like nature of the cells in LMSC-P3, with respect to LMSC-P0. The increased expression of SMA in LMSC-P3, relative to LMSC-P0 could be attributed to the relatively high number of myofibroblastic cells in LMSC-P3 than LMSC-P0.”
Point 2: Please add a mention to the role of ABCB5 in cornea/stem cell markers (lines 121-122).
Response 2: Thank you for pointing this out. The changes are amended in lines 125-126, section 2.2.1 as “ABCB5 plays a vital role in the differentiation of limbal stem cells and is essential for corneal repair.”
Point 3: Please list the sample size used in the appropriate figure legends. Also, specify if the graphs shown are reporting mean +/- stdev or stderror. The statistical test used should also be listed in the figure legend.
Response 3: Thank you for your suggestion. As instructed, all the figure legends are amended (Figures 2, 3, 5, and 7).
Point 4: The authors focus on VEGF and COL5 as genes that are differentially expressed at P3 compared to P0. These findings are unexpected. Can the authors elaborate in the discussion regarding why this might be seen. Have there been comparisons between isolated corneal epithelial cells and keratocytes/fibroblasts in vitro?
Response 4: Thank you for your comment. The elevated levels of VEGFA were indeed observed to be high in LMSC-P3 in our study. This finding was similar to the study by Ma et al. (Invest. Ophthalmol. Vis. Sci. 1999, 40, 1822–1828), and the same is now added to paragraph 8 of the Discussion. This study comparatively studied the angiogenic activity of the epithelial and fibroblast cells of the cornea and limbus, in addition to the conjunctiva.
We haven’t performed comparative studies between the corneal epithelial cells and keratocytes/fibroblasts in vitro. However, our methodology included the comparative study between whole corneal and limbal tissues and the cultures of limbus-derived mesenchymal/stromal stem cells at different stages in culture, the cells of interest in the current study.
Our attempt to get a brief idea of the genes and processes involved in corneal wound healing through the RNA-Seq has revealed COL5 to be exclusively upregulated in the LMSC-P3. At the same time, it was downregulated in the rest of the cells/tissues, i.e., cornea, limbus, and LMSC-P0 (Supplementary table 5 and Figure 5C). Further validation through qRT-PCR has also shown a similar expression pattern (Figure 5D), where COL5 was upregulated in LMSC-P3 and downregulated in LMSC-P0. These findings were similar to the studies by McLaughlin et al (J. Cell Sci. 1989, 94, Pt 2, 371–379,) and DeNigris et al (Connect. Tissue Res. 2016, 57, 1–9) who have shown that the higher number of fibroblasts can elevate the levels of collagen type V. The same has been discussed in lines 519-527; paragraph 5 of Discussion.
Point 5: Minor: Change 'viz' to 'e.g.,'.
Response 5: The ‘viz’ in line 41 is now replaced.
Reviewer 2 Report
This is a very interesting manuscript; however, I would like the authors to address a specific concern about the cytokines and secretory molecules mentioned in line 353. The authors should ensure that sentences with minor punctuation and/or grammatical errors are revised.
Line 63: ICST - The authors should ensure that all abbreviations or acronyms used in the manuscript for the first time are written in full.
Lines 112: The authors should ensure that this sentence with minor spelling error is revised.
Lines 203 - 206: Reword the sentence in order to render it comprehensible.
Lines 250 - 251: “Inflammatory response is down-regulated more strongly in P3 versus P0…” Elaborate on the reason for this in the discussion section.
Lines 273 - 274: Reword the sentence in order to render it comprehensible.
Line 277: "The genes of the cellular processes." It appears that this is a sentence fragment, and as such, it is incomplete.
Lines 353 - 354: Elaborate on these cytokines and secretory molecules.
Lines 393 - 395: This is an incomplete sentence.
Lines 410 - 411: Reword the sentence in order to render it comprehensible.
Lines 415 - 417: “Unlike the earlier studies [71], the positive correlation between the levels of TGF-β and TIMP1 was also not observed in our study.” The authors stated "unlike the earlier studies" but they cited only one reference. Cite the other studies. Explain the reason that there is no correlation between TGF-beta and TIMP-1 in your study.
Author Response
Response to Reviewer 2 Comments
Point 1: This is a very interesting manuscript; however, I would like the authors to address a specific concern about the cytokines and secretory molecules mentioned in line 353. The authors should ensure that sentences with minor punctuation and/or grammatical errors are revised.
Response 1: Thank you for your comments and suggestions. The cytokines and the secretory molecules are discussed in paragraph 2 of the Discussion. All the sentences with typographic/ grammatical errors were revised, as instructed.
Point 2: Line 63: ICST - The authors should ensure that all abbreviations or acronyms used in the manuscript for the first time are written in full.
Response 2: Thank you for pointing out the error. The abbreviation used is expanded as “International Society for Cellular Therapy” and the relevant source is cited (Lines 63-64).
We have also ellobrated other abbreviations.
Point 3: Line 112: The authors should ensure that this sentence with minor spelling errors is revised.
Response 3: The typographic errors are now corrected in (Line 113) and throughout the manuscript.
We again did full proofreading of manucript and corrected as required.
Point 4: Lines 203 - 206: Reword the sentence in order to render it comprehensible.
Response 4: The typographic errors are corrected, and the sentence is modified for a better understanding, as suggested (Lines 208-211) as “The information on the number of genes commonly expressed in one or more cells/tissues was obtained. In addition, the number of genes that were either exclusively up-regulated or exclusively down-regulated in one particular type of cell/tissue was also obtained.”
Point 5: Lines 250 - 251: “Inflammatory response is down-regulated more strongly in P3 versus P0…” Elaborate on the reason for this in the discussion section.
Response 5: Thank you for your suggestion. These findings are elaborated in Lines 430-437; paragraph 4 of the Discussion as “Many studies have proven the anti-inflammatory and immunomodulatory properties of the LMSCs [85–88]. The findings of the current study also support the anti-inflammatory nature of these cells. The overall genes of the inflammatory response (734) were downregulated in LMSC-P3 relative to LMSC-P0 (Figure 5A). The pro-fibrotic gene IL-13 (Figure 7), and inflammatory genes C3, CXCL8 which may lead to corneal neovascularization etc. were downregulated in LMSC-P3. Additionally, the anti-inflammatory gene 1L-10 (Figure 7) was upregulated in LMSC-P3 relative to LMSC-P0.”
Point 6: Lines 273 - 274: Reword the sentence in order to render it comprehensible.
Response 6: As suggested, the sentence is corrected to render it comprehensible (Lines 284-287) as “Genes of epithelial phenotype were found to be strongly down-regulated in LMSC-P3. Genes of the epithelial-to-mesenchymal transition (SNAI1, TWIST1) were down-regulated in both LMSC-P0 and LMSC-P3.”
Point 7: Line 277: "The genes of the cellular processes." It appears that this is a sentence fragment, and as such, it is incomplete.
Response 7: Thank you for bringing this up. A fragment of the sentence got erroneously deleted. The sentence is now wholly written (Lines 292-294) as “In addition to these processes, the GO analysis revealed the DEGs of various signaling pathways such Wnt, TGF-β and stem cell pathways (Supplementary Figure 3 and Supplementary Table 6).”
Point 8: Lines 353 - 354: Elaborate on these cytokines and secretory molecules.
Response 8: The cytokines and the secretory molecules are discussed as suggested, Lines 371-387; in paragraph 1 of Discussion as
“IL-1 and its isoforms (IL-1α and IL-1β) produced by the epithelial cells during corneal in-jury, promote the production of TNF-α, KGF, and HGF [58,59]. Together with TNF-α, IL-1 also modulates the production of growth factors (PDGF and family) that modulate the chemotaxis and proliferation of corneal fibroblasts[60]. They also enhance the levels of cytokines such as G-CSF, neutrophil-activating peptide, IL-3 precursor, IL-4, IL-6, IL-7, IL-8, IL-9, and IL-17 [61]. These cytokines trigger the entry of inflammatory cells to the site of injury [62,63]. HGF and KGF released by the stromal fibroblasts, along with bFGF, IGF, and EGF, modulate the interactions between epithelial and stromal cells, regulating the migration and differentiation of damaged epithelial cells [64–68]. IL-6, a multifunctional cytokine, modulates the repair of the cornea in many ways. It enhances the epithelial wound closure and low levels of IL-6 delays the healing [69–71]. It reduces the levels of IL-1 and TNF- α, lowering the inflammation[72]. A study by Samaeekia et al [73]has shown that the exosomes isolated from the corneal and peripheral limbal MSCs enhance the migration and proliferation of corneal epithelial cells in vitro. The co-culture of corneal epithelial cells and corneal stromal cells has been shown to reduce the levels of pro-inflammatory cytokines and enhance the number of viable epithelial cells following an injury [74].”
Point 9: Lines 393 - 395: This is an incomplete sentence.
Response 9: Thank you for bringing this up. A fragment of the sentence got erroneously deleted. The sentence is now wholly written in Line 430 as
“Genes belonging to signaling pathways such as Wnt (211 DEGs), TGF-β (85 DEGs), stem cell (23 DEGs) and wound healing pathways (254 DEGs) were also obtained”
Point 10: Lines 410 - 411: Reword the sentence in order to render it comprehensible.
Response 10: The sentence is now wholly written, mentioning the genes’ families in Line 465 as “These two groups of genes i.e. the MMP family and the TIMP family, also a play vital role in the development of cornea.”
Point 11: Lines 415 - 417: “Unlike the earlier studies [71], the positive correlation between the levels of TGF-β and TIMP1 was also not observed in our study.” The authors stated "unlike the earlier studies" but they cited only one reference. Cite the other studies. Explain the reason that there is no correlation between TGF-beta and TIMP-1 in your study.
Response 11: Thank you for your suggestion. The studies that showed a positive correlation between TGF-β and TIMP1 are cited. However, we did not find any study where the levels of TGF-β and TIMP1 were studied in normal conditions or cultures. In our study, we quantified the levels of these genes in normal conditions where no induction of TGF-β or TIMP1 was done to study their correlation. This observation of no correlation which is contradictory to other studies of disease conditions is already discussed in paragraph 6 of the Discussion (Line 470) with relevant references doi: 10.1158/1541-7786.MCR-05-0140; doi: 10.1371/journal.pone.0057474.
Reviewer 3 Report
The authors were focused on transcriptomic profiling of human limbus-derived stromal/mesenchymal stem cells. The topic is of interest since to elucidate the process of wound healing including corneal wound is still a challenge.
I think the manuscript includes sufficient methods based on molecular biology. I do not have any critical comments. Only one question: How many human donor corneas were analyzed?
Author Response
Thank you for your comment. In this study, 21 donor corneas were utilized for RNA-Seq, qRT-PCR, and Immunostaining. The same has been mentioned now in section 5.1 of Materials and Methods (Lines 536-537).
Round 2
Reviewer 2 Report
The authors have completed a good revision of the original manuscript.
Author Response
Thank you